# Stem Cell-Based Regenerative Therapy and Derived Products in COPD: A Systematic Review and Meta-Analysis

**DOI:** 10.3390/cells11111797

**Published:** 2022-05-30

**Authors:** Luigino Calzetta, Marina Aiello, Annalisa Frizzelli, Francesca Camardelli, Mario Cazzola, Paola Rogliani, Alfredo Chetta

**Affiliations:** 1Respiratory Disease and Lung Function Unit, Department of Medicine and Surgery, University of Parma, 43126 Parma, Italy; marina.aiello@unipr.it (M.A.); annalisa.frizzelli@unipr.it (A.F.); alfredoantonio.chetta@unipr.it (A.C.); 2Unit of Respiratory Medicine, Department of Experimental Medicine, University of Rome “Tor Vergata”, 00133 Rome, Italy; francesca.camardelli@gmail.com (F.C.); mario.cazzola@uniroma2.it (M.C.); paola.rogliani@uniroma2.it (P.R.)

**Keywords:** COPD, mesenchymal stem cells, meta-analysis, regenerative

## Abstract

COPD is an incurable disorder, characterized by a progressive alveolar tissue destruction and defective mechanisms of repair and defense leading to emphysema. Currently, treatment for COPD is exclusively symptomatic; therefore, stem cell-based therapies represent a promising therapeutic approach to regenerate damaged structures of the respiratory system and restore lung function. The aim of this study was to provide a quantitative synthesis of the efficacy profile of stem cell-based regenerative therapies and derived products in COPD patients. A systematic review and meta-analysis was performed according to PRISMA-P. Data from 371 COPD patients were extracted from 11 studies. Active treatments elicited a strong tendency towards significance in FEV_1_ improvement (+71 mL 95% CI -2–145; *p* = 0.056) and significantly increased 6MWT (52 m 95% CI 18–87; *p* < 0.05) vs. baseline or control. Active treatments did not reduce the risk of hospitalization due to acute exacerbations (RR 0.77 95% CI 0.40–1.49; *p* > 0.05). This study suggests that stem cell-based regenerative therapies and derived products may be effective to treat COPD patients, but the current evidence comes from small clinical trials. Large and well-designed randomized controlled trials are needed to really quantify the beneficial impact of stem cell-based regenerative therapy and derived products in COPD.

## 1. Introduction

Chronic obstructive pulmonary disease (COPD) is a common, preventable, and treatable respiratory disease, characterized by persistent respiratory symptoms and airflow re-striction [1]. Based on the recent World Health Organization’s Global Health Estimates, COPD is currently the third leading cause of mortality worldwide, with 3.2 million deaths reported in 2019 [2].

Chronic inflammation, protease–antiprotease imbalance, and oxidative stress are regarded as the pathogenic triad of COPD [3]. Prolonged exposure to noxious agents and air pollutants including tobacco smoke, induces an enhanced innate immune response, leading to the recruitment of neutrophils and activated macrophages into the lungs, mucus secretion, and epithelial cells activation [4]. Dendritic cells promote the subsequent adaptive immune responses mediated by CD4+ T helper cells and CD8+ cytotoxic T cells, as well as B lymphocytes, that result in the development of lymphoid follicles on chronic inflammation [4,5]. Neutrophils, macrophages, and CD8+ cytotoxic T cells are primed to release a range of proteolytic enzymes, namely serine-, metallo-, and cysteine-proteases, and to inactivate several antiproteases by oxidation [6]. This will result in an excessive and unregulated proteolytic cleavage and degradation of extracellular matrix components, generating fragments of elastin and collagen fibers that have chemotactic activity for monocytes [7]. Progressive alveolar tissue destruction and defective repair and defense mechanisms lead to conditions of emphysema and small airway fibrosis, along with gas trapping and airflow obstruction [1].

Currently, available treatments are primarily directed at reducing symptoms of COPD and prevent the risk of future exacerbations [1]. Inhaled bronchodilators acting via different mechanisms of action remain the pharmacological mainstay for treating most COPD patients, either administered alone or in combination as dual therapy or as triple therapy with the addition of an inhaled corticosteroid (ICS) [1,8,9].

Findings from a recent systematic review and post hoc analysis [10] showed that current pharmacotherapy for COPD improved the rate of forced expiratory volume in the 1st second (FEV_1_) decline to a small extent compared to placebo. In this respect, long-acting bronchodilator- and ICS-containing treatments reduced the decline of FEV_1_ by 4.9 mL/year and 4.7 mL/year, respectively [10]. Nevertheless, none of these pharmacological treatments interfere with the progressive nature of the disease [11].

Considering that no curative treatment for COPD is available, innovative therapeutic approaches such as regenerative therapy have been proposed and investigated [12], with the goal of repairing or regenerating damaged functional structures of the respiratory system and improving functionality [13].

Stem cell-based regenerative therapy is regarded as any treatment for a disease or a medical condition based on the use of any type of viable human stem cells, namely, embryonic stem cells, induced pluripotent stem cells, and adult stem cells for autologous and allogeneic infusion [14]. Mesenchymal stem cells (MSC) are among the most studied stem cells and their therapeutic potential has been extensively demonstrated in numerous pre-clinical models of pulmonary disorders [15,16,17,18,19,20]. Interestingly, the beneficial effects of MSC have been mostly ascribed to the “secretome”, an MSC-derived molecule containing an extended range of bioactive molecules, including cytokines, chemokines, growth factors, angiogenic factors, and extracellular vesicles, that potentially modulate regenerative activity in a paracrine manner, as demonstrated in preclinical studies [21].

Stem cell-based regenerative therapy is relatively new to the field of COPD and the current Global Initiative for COPD (GOLD) document does not provide any information concerning this innovative therapeutic approach [1].

Considering that no quantitative synthesis of the clinical benefit of stem cell-based regenerative therapies and derived products for COPD is currently available, the aim of this study was to provide a systematic review and meta-analysis on the efficacy profile of stem cell-based regenerative therapies and derived products in COPD patients. 

## 2. Materials and Methods

### 2.1. Search Strategy and Study Eligibility

The protocol of this qualitative and quantitative synthesis has been registered in the international prospective register of systematic reviews (PROSPERO, registration ID: CRD42022093446) and performed in agreement with the Preferred Reporting Items for Systematic Reviews and Meta-Analyses Protocols (PRISMA-P) [22]. The relative flow diagram is shown in Figure 1. This study satisfied all the recommended items reported by the PRISMA 2020 checklist (Appendix A) [23].

A comprehensive literature search was performed for clinical studies evaluating the efficacy profile of stem cell-based regenerative therapies and derived products for the treatment of COPD. In this regard, the PICO (Patient problem, Intervention, Comparison, and Outcome) framework was applied to develop the literature search strategy, as previously reported [24]. Namely, the “Patient problem” included patients suffering from COPD; the “Intervention” regarded the administration of stem cell-based regenerative therapy and derived products; the “Comparison” was performed with respect to baseline or control (CTL); the assessed “Outcomes” were lung function, acute exacerbation of COPD (AECOPD), hospitalization due to AECOPD, exercise capacity, and blood gas analysis.

The search was performed in ClinicalTrials.gov, accessed on 24 May 2022, Cochrane Central Register of Controlled Trials (CENTRAL), Embase, EU Clinical Trials Register, MEDLINE, Scopus, and Web of Science, in order to provide for relevant studies published up to 24 May 2022. The summary of the search string was as follows: (regenerative OR (Mesenchymal Stromal Cells) OR adipose) AND therapy AND (COPD OR emphysema). Detailed information regarding the expanded search string and translations are reported in Appendix A. Citations of previous published reviews were checked to select further pertinent studies, if any [25].

Literature search results were uploaded to Eppi-Reviewer 4 (EPPI-Centre Software, version 4.12.5.0, London, UK), a web-based software program for managing and analyzing data in literature reviews that facilitates collaboration among reviewers during the study selection process.

### 2.2. Study Selection

Published interventional and observational clinical studies on COPD patients that reported data concerning the efficacy of stem cell-based regenerative therapies and derived products vs. baseline or CTL were included in the qualitative and quantitative syntheses. Two reviewers independently examined the studies, and any difference in opinion concerning the selection of relevant studies from literature searches and databases was resolved by consensus.

### 2.3. Data Extraction

Data from the included clinical studies were extracted from published papers, Appendix A, and the public database ClinicalTrials.gov, accessed on 5 April 2022. Data were checked for study characteristics and duration, number of analyzed patients, type of stem cell-based treatment or derived products with doses of medications and regimen of administration, main inclusion criteria, age, gender, FEV_1_, smoking habit, investigated outcomes, Jadad Score [26], Newcastle–Ottawa Scale (NOS) score, and the Cochrane risk of bias (RoB) [27].

Data were extracted in agreement with Data Extraction for Complex Meta-anALysis (DECiMAL) recommendations [28]. When necessary, arithmetic mean and standard deviation were estimated from the median, interquartile range (IQR), range, and the sample size, as previously described [29].

The inter- and intra-rater reliability for data abstraction was assessed via the Cohen’s Kappa score, as previously described [30]. Briefly, Cohen’s Kappa ≥ 0.80 indicated excellent agreement, coefficients between 0.61 and 0.80 represented substantial agreement, coefficients between 0.41 and 0.61 moderate agreement and <0.41 fair to poor agreement. 

### 2.4. Endpoints

The endpoint of this pairwise meta-analysis was to assess the efficacy profile of different stem cell-based regenerative therapies and derived products vs. baseline or CTL in terms of changes in FEV_1_, forced vital capacity (FVC), residual volume (RV), diffusing capacity for carbon monoxide (DLCO), risk of AECOPD, risk of hospitalization due to AECOPD, changes in 6 min walking test (6MWT) and partial pressure of carbon dioxide (pCO2).

### 2.5. Data Synthesis and Analysis

A pairwise meta-analysis was performed to quantify the efficacy of stem cell-based regenerative therapies and derived products compared to baseline or CTL. Results were expressed as mean difference (MD), as relative risk (RR) according to the analyzed variables, and 95% confidence interval (95% CI). When outcomes were reported in the studies by using different metrics, then results were expressed as standardized mean difference (SMD = [difference in mean outcome between groups] × [standard deviation of outcome among participants] − 1).

Since data were selected from a series of studies performed by researchers operating independently and a common effect size could not be assumed, binary random-effects model was used to balance the study weights and adequately estimate the 95% CI of the mean distribution of stem cell-based therapies and derived products’ effect on the investigated variables [31,32,33,34].

When ≥3 studies reported lung function outcomes expressed as volume, subgroup analyses were performed, and the overall effect estimate calculated as MD and 95% CI. A further subgroup analysis was conducted by including exclusively randomized controlled trials (RCTs) reporting data on FEV_1_.

The test for heterogeneity (I^2^) was performed to quantify the between-study dissimilarity, as previously reported [35], and sensitivity analysis was carried out to identify the studies that introduced substantial levels of heterogeneity (I^2^ > 50%) in the quantitative synthesis [36]. In the sensitivity analysis, once significant and/or substantial heterogeneity was resolved, fixed-effect (Mantel–Haenszel) method was used if sparse data were reported, as previously described [37]. A further sensitivity analysis was undertaken after selectively excluding observational studies as a potential source of bias. 

### 2.6. Quality of Studies, Risk Bias, and Evidence Profile

The summary of the risk of bias for each included RCT was analyzed via the Jadad score [26] and Cochrane RoB 2 [27]. The Jadad score ranges from 1 to 5 (score of 5 being the best score), and the quality of studies was ranked as follows: score < 3, low quality; score = 3, medium quality; score > 3 high quality. The weighted assessment of the risk of bias was analyzed via the Cochrane RoB 2 [27].

The NOS was used to assess the quality of observational cohort studies [38]. According to NOS, a study can be awarded with a maximum of one star for each item within the “Selection” and “Outcome” categories, and a maximum of two stars can be given for “Comparability” [38]. In the present quantitative analysis, the NOS quality assessment score was established to be in the range between zero and a maximum of nine stars. Studies reporting a NOS score ≥ 7 were considered of high quality, whereas those reporting a NOS score ≤ 6 were considered of low quality. For the NOS category “Outcome”, a follow-up period of at least ≃ 6 months was considered adequate to obtain the outcomes of interest from the included studies [39].

Funnel plot and Egger’s test were performed to assess the origin and risk of publication bias if ≥10 studies were included in the meta-analysis [40], and the following regression equation was applied: SND = a + b × precision, where SND represents the standard normal deviation (treatment effect divided by its standard error (SE)), and precision represents the reciprocal of the standard error. Evidence of asymmetry from Egger’s test was considered to be significant at *p* < 0.1, and the graphical representation of 90% confidence bands reported [41,42,43].

The quality of the evidence was assessed for the main endpoint in agreement with the Grading of Recommendations Assessment, Development, and Evaluation (GRADE) system, indicating ++++ for high quality of evidence, +++ for moderate quality of evidence, ++ for low quality of evidence, and + for very low quality of evidence [27]. Two reviewers independently assessed the quality of studies, risk bias, and evidence profile, and any difference in opinion was resolved by consensus.

### 2.7. Software and Statistical Significance

GraphReader was used to extract data from the figures, when necessary (graphreader.com accessed on 24 May 2022), OpenMeta-Analyst (version 12.11.14, Wallace et al., Tufts University, Boston, MA, USA) [35] software was used to perform the pairwise meta-analysis, GraphPad Prism (version 7.0a, GraphPad Software Inc., San Diego, CA, USA) software to graph the data, GRADEpro GDT (online version available from gradepro.org, McMaster University and Evidence Prime Inc., Hamilton, ON, Canada) to assess the quality of evidence [27], and the robvis visualization software (online version available from mcguinlu.shinyapps.io/robvis/, McGuinness et al., University of Bristol, Bristol, UK) to perform the RoB 2 tool [44,45]. The statistical significance of the effect estimates resulting from the pairwise meta-analysis was assessed for *p* < 0.05.

## 3. Results

### 3.1. Study Characteristics

Of the 415 potentially relevant records identified in the initial search, 14 studies [46,47,48,49,50,51,52,53,54,55,56,57,58,59] were deemed eligible for a qualitative synthesis. Since 3 studies [48,49,50] reported non-extractable data on efficacy outcomes, the quantitative synthesis (pairwise meta-analysis) was carried out on 11 studies [46,47,51,52,53,54,55,56,57,58,59]. The effect estimates of the pairwise meta-analysis were obtained from 371 COPD patients selected from 6 non-randomized, non-controlled clinical trials [46,47,52,53,54,55], 3 RCTs [56,57,58], and 2 observational cohort studies [51,59]. The relevant patient demographics, study characteristics, and Jadad score have been summarized in Table 1.

The investigated stem cell-based regenerative therapies included human adipose tissue-derived stem cells (ADSC), bone marrow (BM) mononuclear cells (BMMC), BM-derived MSC (BM-MSC), peripheral blood stem cells (PBSC), and umbilical cord MSC (UC-MSC). One study [46] evaluated the therapeutic effect of a newly engineered soluble product obtained from placenta-derived MSC (PL-MSC), containing a high concentration of immunosuppressive factors including soluble tumor necrosis factor (TNF) receptors I and II, interleukin (IL)−1 receptor antagonist, and soluble receptor for advanced glycation end products.

The inter-rater reliability for data abstraction was excellent before and after the learning process (Cohen’s Kappa 0.96 and 1.00, respectively). The intra-rater reliability produced a Cohen’s Kappa of 1.00 after the learning process.

### 3.2. Pairwise Meta-Analysis

#### 3.2.1. Lung Function

Treatment with stem cell-based regenerative therapies and derived products resulted in a strong tendency towards a significant (*p* = 0.057) improvement in FEV_1_ compared to baseline or CTL (SMD 0.67, 95% CI -0.02–1.36; I^2^ 89.65%; GRADE +) (Figure 2A). The sensitivity analysis indicated that after excluding the study by Harrell et al. [46], the substantial level of heterogeneity was resolved (I^2^ 0%), but no significant (*p* > 0.05) difference was detected with respect to FEV_1_ compared to baseline or CTL (Appendix A). In the subgroup analysis for FEV_1_ expressed as volume, a strong tendency towards significance (*p* = 0.056) was observed for FEV_1_ vs. baseline or CTL (MD 71.45 mL, 95% CI -1.91–144.81; I^2^ 0%; GRADE +++) (Figure 2A’). A further subgroup analysis carried out by including only RCTs did not show a significant (*p* > 0.05) change in FEV_1_ vs. CTL (Appendix A).

Stem cell-based regenerative therapies did not significantly (*p* > 0.05) improve FVC compared to baseline or CTL (SMD 0.31, 95% CI -0.33–0.95; I^2^ 46.88%; GRADE ++) (Figure 2B). Subgroup analysis did not show a significant (*p* > 0.05) increase in FVC expressed in mL when comparing active treatments with baseline or CTL (MD 283.50 mL, 95% CI -211.10–778.09; I^2^ 78.80%; GRADE +) (Figure 2B’). The study by Harrell et al. [46] investigating the effects of an MSC-derived product did not report data on FVC.

Stem cell-based regenerative therapies did not induce a significant (*p* > 0.05) reduction in RV compared to baseline or CTL (SMD -0.43, 95% CI -1.69–0.83; I^2^ 77.94%; GRADE +) (Figure 2C). The sensitivity analysis indicated that the main source of the substantial heterogeneity affecting the effect estimate of RV was introduced by the study of Stolk et al. [53] (I^2^ 0%). When substantial heterogeneity was resolved by sensitivity analysis, no significant (*p* > 0.05) difference was detected for the reduction in RV vs. baseline or CTL (Appendix A). The study by Harrell et al. [46] investigating the effects of an MSC-derived product did not report data on RV.

No significant (*p* > 0.05) change in DLCO was found between treatment with stem cell-based regenerative therapies and derived products and CTL (SMD 0.32, 95% CI -0.29–0.93; I^2^ 34.75%; GRADE +++) (Figure 2D). The study by Harrell et al. [46] investigating the effects of an MSC-derived product did not report data on DLCO.

#### 3.2.2. AECOPD

The pairwise meta-analysis for the risk of AECOPD was not performed since only the study by Le Thi Bich et al. [47] reported efficacy data on the number of AECOPD.

#### 3.2.3. Hospitalization Due to AECOPD

Stem cell-based regenerative therapies did not significantly (*p* > 0.05) modulate the risk of hospitalization due to AECOPD compared to baseline or CTL (RR 0.77, 95% CI 0.40–1.49; I^2^ 0%; GRADE ++) (Figure 3A). The study by Harrell et al. [46] investigating the effects of an MSC-derived product did not report data on hospitalizations due to AECOPD.

#### 3.2.4. Exercise Capacity

Patients treated with stem cell-based therapies and derived products showed a significant (*p* < 0.01) improvement in the 6MWT compared to baseline or CTL (MD 52.63 m, 95% CI 18.42–86.83; I^2^ 78.26%; GRADE ++) (Figure 3B). Sensitivity analysis performed by excluding the study by Le Thi Bich et al. [47] confirmed a significant (*p* < 0.001) increase in the 6MWT vs. baseline or CTL (MD 69.21 m, 95% CI 66.71–71.72; GRADE +++) and resolved the substantial level of heterogeneity (I^2^ 0%) (Appendix A).

#### 3.2.5. Blood Gas Analysis

Stem cell-based regenerative therapies did not significantly (*p* > 0.05) modulate pCO2 compared to baseline or CTL (MD 9.57 mmHg, 95% CI -11.08–30.22; I^2^ 98.51%; GRADE +) (Figure 3C). Sensitivity analysis indicated that after excluding the study by Ribeiro-Paes et al. [54], heterogeneity was resolved (I^2^ 0%), but no significant difference in the effect estimate of pCO2 was detected vs. CTL (Appendix A). The study by Harrell et al. [46] investigating the effects of an MSC-derived product did not report data on pCO2.

### 3.3. Risk of Bias and Quality of Evidence

All the RCTs [56,57,58] included in the pairwise meta-analysis were ranked as being of medium- to high-quality, in accordance with the Jadad score. Two studies [56,57] were of medium quality (Jadad score = 3) and one [58] was of high quality (Jadad > 3). The traffic light plot for the assessment of each included RCT is reported in Figure 4A and the weighted plot for the assessment of the overall risk of bias by domains is shown in Figure 4B. All the RCTs had a low risk of bias for the randomization process (3 (100.0%)), missing outcome data (3 (100.0%)), and selection of the reported results (3 (100.0%)). All 3 RCTs had some concerns in the domain of measurement of the outcomes, 2 had some concerns in the domain of deviations from intended intervention, and 1 in the randomization process. Overall, 7 studies [46,47,51,52,53,54,55,59] could not be ranked via the Cochrane RoB2 and Jadad score, either because they were non-randomized and non-controlled [46,47,52,53,54,55], or due to the observational design of the study [51,59]. The quality of evidence resulting from the observational cohort studies [51,59] was considered as low, with a NOS score ≤ 6.

In agreement with the recommendation [40] for testing funnel plot asymmetry, only the meta-analysis for the impact of stem cell-based therapies and derived products on FEV_1_ vs. baseline or CTL was eligible for publication bias detection. The visual inspection of funnel plot reported asymmetry and dispersion (Figure 5A), although the Egger’s test indicated that the results were not affected by significant (*p* = 0.670) publication bias (Figure 5B). The sensitivity analysis performed to resolve heterogeneity by excluding the study of Harrell et al. [46] reduced the visual asymmetry and dispersion and confirmed the lack of significant (*p* = 0.669) publication bias (Figure 5C,D).

## 4. Discussion

The main findings of this study indicated that treatment with stem cell-based regenerative therapies and derived products significantly improved exercise capacity in COPD patients and produced a trend towards significance for an improvement in FEV_1_, but no other lung function parameters showed a significant difference compared to baseline or CTL. Stem cell-based regenerative therapies neither reduced the risk of hospitalization due to AECOPD nor modulated the level of pCO2. Sensitivity analysis confirmed the improvement of exercise capacity in treated patients and did not report any other significant difference in the effect estimates. The subgroup analysis detected a strong tendency towards a significant improvement in FEV_1_ compared to baseline or CTL, while FVC increased not significantly. The results of this meta-analysis are characterized by a very-low-to-moderate quality of evidence and are not affected by significant publication bias, despite a certain level of asymmetry and dispersion observed via funnel plot analysis.

Many preclinical studies have yielded positive results supporting the hypothesis that stem cell-based regenerative therapies may have a therapeutic benefit in COPD, and pro-vided an important basis for further clinical investigations with different sources of MSC in COPD patients [13]. In this regard, a previous meta-analysis [60] of animal studies, mainly rats and mice, confirmed that MSC-based therapies enhance lung tissue repair, improve lung function, and reduce inflammation.

Nevertheless, although preliminary clinical findings demonstrated no remarkable safety issues with stem cell-based regenerative therapies, as yet they have not established definitive therapeutic effects in COPD patients [61,62]. In this respect, the advantage of conducting a meta-analysis lies in its ability to combine data from separate and relatively small studies that may have been underpowered to detect statistically significant differences between one intervention and another [63]. To the best of our knowledge, this is the first quantitative synthesis to have investigated the efficacy profile of stem cell-based regenerative therapies and derived products in COPD patients.

Several methodological issues may have hampered the translation of preclinical evidence to the clinic. Firstly, pre-clinical studies commonly used animal models to just mimic mild to moderate stages of COPD via acute or sub-acute phase of lung tissue damage [13,60]. By contrast, the clinical studies conducted so far have prevalently recruited patients in an advanced stage of the disease, mostly moderate to severe COPD with a chronic inflammatory condition [47,49,52,53,54,56,57,58], therefore lung tissue injuries might have been too severe to be reversed by infusion of stem cells [58]. Secondly, no scientific consensus has been established concerning the number of stem cells present in each dose, treatment schedules, timing of administration, and stem cells source [13]. In most studies, the original stem cell dose infused to COPD patients depended on body’s weight such as 1.0–2.0 × 10^6^ cells/kg [47,48,52,53,55], while Weiss et al. [58] empirically established dosing on data from MSC trials for other diseases [64,65] that may not be effective in chronic lung disorders.

Heterogeneity of stem cell populations is another obstacle that may affect clinical outcomes. In rodent models of emphysema, MSC have been isolated from several adult tissues including adipose tissue [20,61,66,67,68], umbilical cord [69], bone marrow [20,70,71,72,73], placental tissue [74], and lung [20,75]. Based on the site of origin, MSC may display different phenotypes that result in changes in the immunomodulatory, anti-inflammatory, and regenerative effects, as well as in culture expansion [76,77].

An increasing body of evidence indicates that the therapeutic benefit of exogenously infused MSC is mainly a consequence of their secretory properties [78]. Considering that the full therapeutic potential of transplanted MSC is limited by the inability to sufficiently migrate and engraft into target tissue due to culture conditions, donors’ gender and age, and delivery methods [79], attention has recently shifted toward the MSC-derived product secretome, which is currently considered a potential replacement for MSC-based therapy [80], with several biological and logistical advantages. Harnessing the MSC-derived secretome permits us to avoid problems that could be encountered with infused MSC, including the unsettled cell differentiation into undesired tissues in response to local growth factors [81,82]. In addition, MSC need to be expanded in culture to reach an optimal number for transplantation, whereas MSC-derived secretome is readily available for the treatment of acute conditions [83].

Currently, 33 clinical trials of stem cell-based regenerative therapies for COPD are registered at ClinicalTrials.gov, accessed on 24 May 2022, of which 12 studies are still ongoing (Table 2). Many of them lack the approval by the appropriate national regulatory authorities or are pay-to-participate trials of unclear scientific validity [25]. Considering that stem cell-based regenerative therapies are not approved for the treatment of COPD and that stem cell tourism has become an emerging global problem [84], international organizations such as the International Society for Stem Cell Research have taken strong actions to promote rigor and transparency against unregulated stem cell-based interventions [85].

The limitations of this meta-analysis are mainly related with the intrinsic weaknesses of the included studies, characterized by differences in the study design, follow up duration, treatment regimen, and in the reporting of the results. The included clinical studies were prevalently based on a small sample size, thus were potentially underpowered to determine a beneficial response of efficacy. In the present analysis, although data on FEV_1_ failed to meet the significance threshold of *p* < 0.05 and rather showed a trend towards significance, this may be an indication that the study was not able to demonstrate a difference, perhaps due to insufficient sample sizes [86]. In this respect, categorizing a continuous variable such as *p*-value as statistically significant or not-significant is misleading, and rather it should be interpreted in the light of its context, by assessing whether what has been detected is clinically relevant [87,88]. Moreover, most clinical trials did not use a CTL as comparator. Additionally, the results of the present study cannot be extended to a general population of COPD patients, as subjects enrolled in the included studies were mostly affected by moderate to severe COPD with pulmonary emphysema.

Of note, although FEV_1_ and 6MWT are commonly used to assess lung function and exercise capacity in patients with COPD, these clinical endpoints are characterized by a certain level of between-test variability that may affect reproducibility [89,90,91,92]. Therefore, considering that in COPD, the tissue destruction affects almost all structural components of the lungs [93,94], indeed the efficacy outcomes related to the cell-based regenerative therapy and derived products should be assessed also at cellular and biochemical level, by taking into account for sensitive and specific biomarkers such as exosomes and microvesicles [95].

Concluding, the use of stem cell-based therapies and derived products offers a considerable therapeutic potential in regenerative medicine. Large-scale trials are needed to explore treatment effects across all stages of COPD severity by applying standardized protocols and obtain datasets that consistently support moving to Phase III RCTs and provide evidence that can give conclusions regarding clinical efficacy.

## Figures and Tables

**Figure 1 cells-11-01797-f001:**
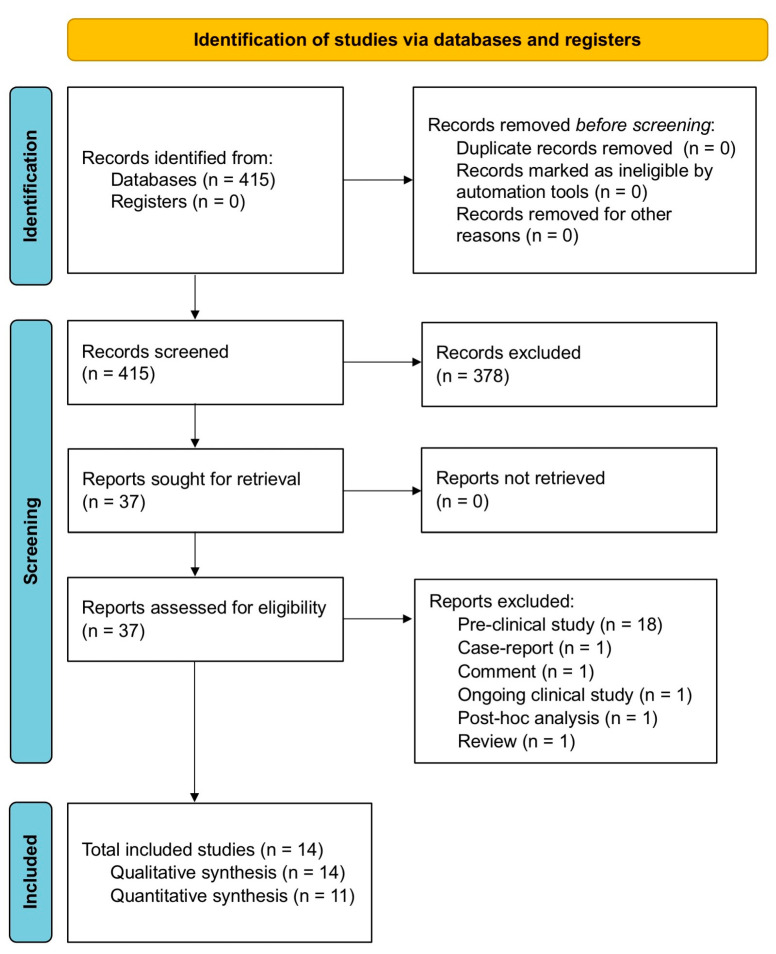
PRISMA 2020 flow diagram for the identification of the clinical studies included in the qualitative and quantitative synthesis. PRISMA: Preferred Reporting Items for Systematic Reviews and Meta-Analyses.

**Figure 2 cells-11-01797-f002:**
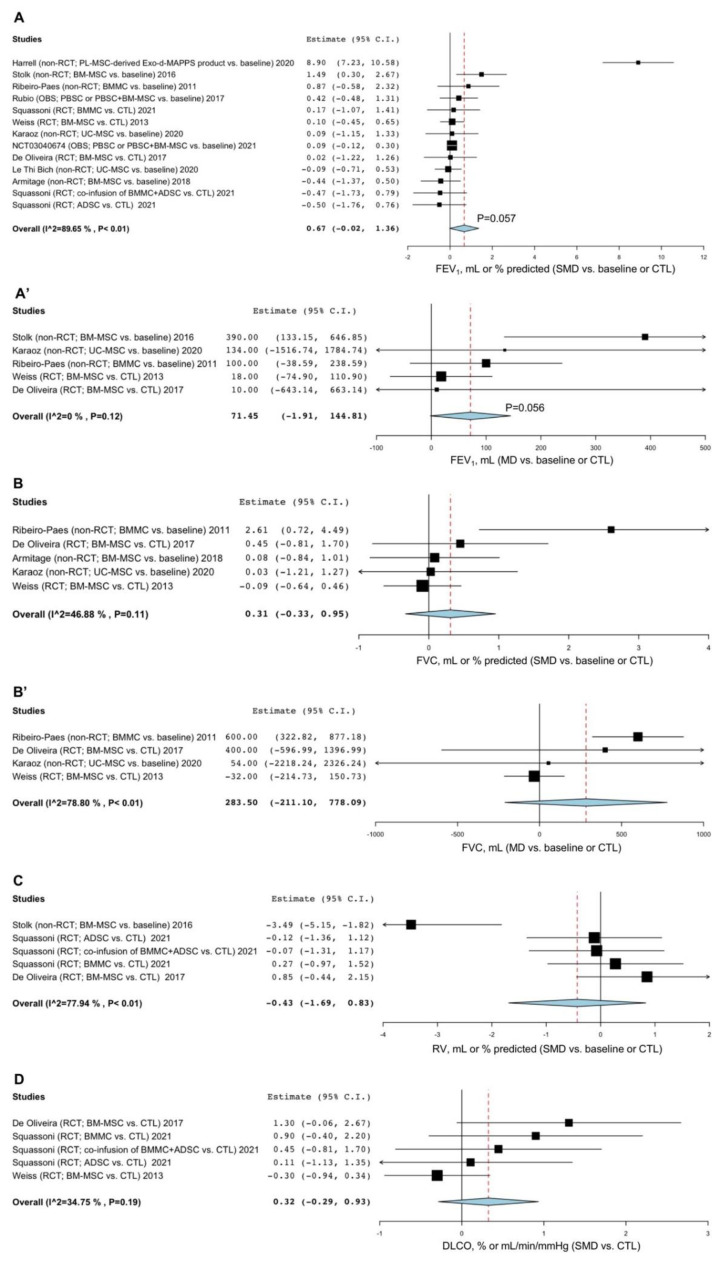
Forest plots of the impact of stem cell-based regenerative therapies vs. baseline or CTL on FEV_1_ (**A**), FVC (**B**), RV (**C**), and DLCO (**D**) and subgroup analysis on the MD in FEV_1_ (**A**’) and FVC (**B**’) reported as volume in mL. ADSC: adipose tissue-derived stem cells; BMMC: bone marrow mononuclear cells; BM-MSC: bone marrow-derived mesenchymal stem cells; CTL: control; DLCO: diffusing capacity for carbon monoxide; FEV_1_: forced expiratory volume in the 1st second; FVC: forced vital capacity; MD: mean difference; OBS: observational study; PBSC: peripheral blood stem cells; PL-MSC: placenta-derived mesenchymal stem cells; RCT: randomized controlled trial; RV: residual volume; SMD: standardized mean difference; UC-MSC: umbilical cord mesenchymal stem cells.

**Figure 3 cells-11-01797-f003:**
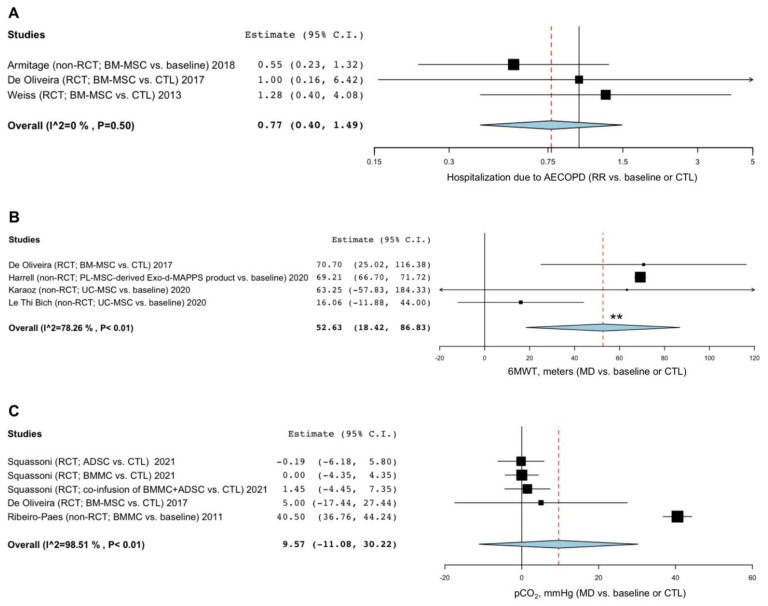
Forest plots of the impact of stem cell-based regenerative therapies vs. baseline or CTL on the risk of hospitalization due to AECOPD (**A**), 6MWT (**B**), and pCO2 (**C**). ADSC: adipose tissue-derived stem cells; AECOPD: acute exacerbation of COPD; BMMC: bone marrow mononuclear cells; BM-MSC: bone marrow-derived mesenchymal stem cells; COPD: chronic obstructive pulmonary disease; CTL: control; MD: mean difference; pCO2: partial pressure of carbon dioxide; PL-MSC: placenta-derived mesenchymal stem cells; RCT: randomized controlled trial; RR: relative risk; UC-MSC: umbilical cord mesenchymal stem cells; 6MWT: 6 min walking test. ** indicates *p*-value < 0.01.

**Figure 4 cells-11-01797-f004:**
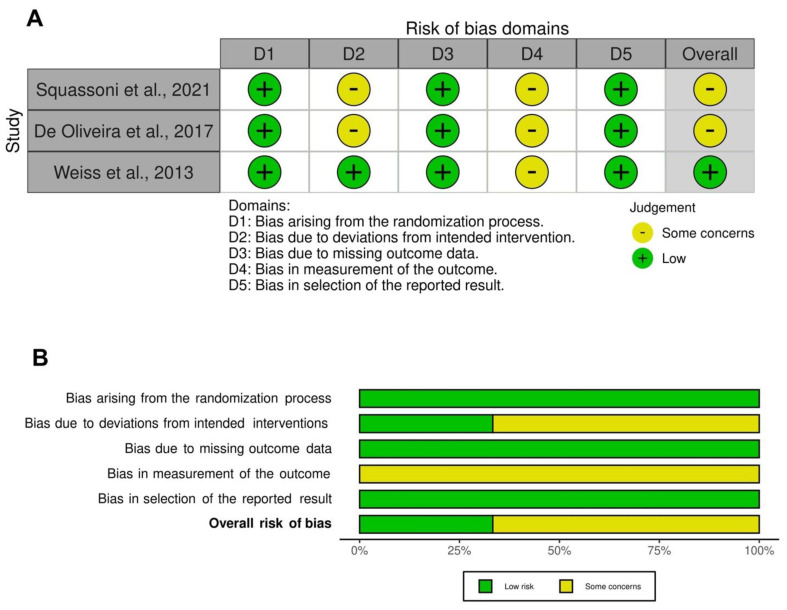
Assessment of the risk of bias via the traffic light plot of each included RCT (**A**) and the weighted plot for the assessment of the overall risk of bias via the Cochrane RoB 2 tool (**B**) (*n* = 3 studies). Traffic light plot reports five risk of bias domains: D1, bias arising from the randomization process; D2, bias due to deviations from intended intervention; D3, bias due to missing outcome data; D4, bias in measurement of the outcome; D5, bias in selection of the reported result; yellow circle indicates some concerns on the risk of bias and green circle represents low risk of bias. RCT: randomized controlled trial; RoB: risk of bias [56,57,58].

**Figure 5 cells-11-01797-f005:**
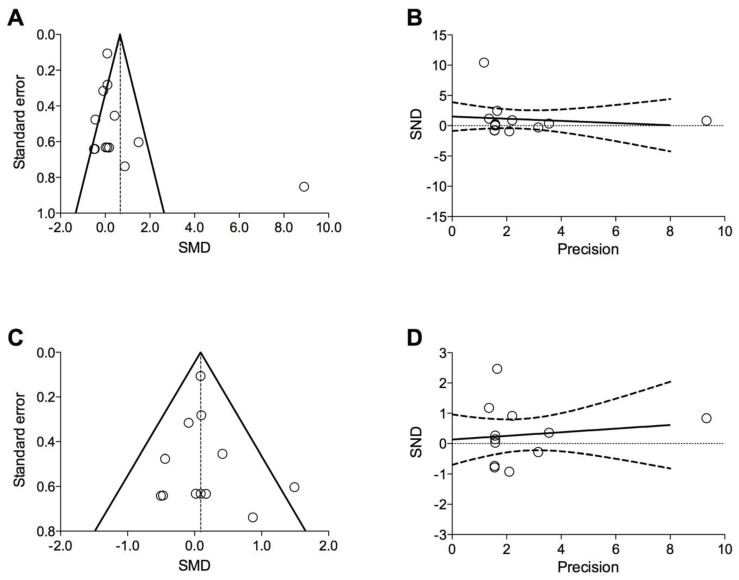
Funnel plot (**A**,**C**) and graphical representation of Egger’s test (**B**,**D**) for the overall impact of stem cell-based regenerative therapies and derived products vs. baseline or CTL on FEV_1_ (**A**,**B**) and for the sensitivity analysis on FEV_1_ (**C**,**D**). CTL: control; FEV_1_: forced expiratory volume in the 1st second; SMD: standardized mean difference; SND: standard normal deviation.

**Table 1 cells-11-01797-t001:** Study characteristics of the studies included in the qualitative and/or quantitative synthesis.

Author, Year, and References	Trial Number Identifier	Study Characteristics	Study Duration (Weeks)	Number of Analyzed Patients	Types of Stem Cell-Based Treatment or Derived Products	Regimen and Route of Administration	Patients’ Characteristics	Age (Years)	Male (%)	Pre-Bronchodilator FEV_1_ (% Predicted)	Current Smokers (%)	Smoking History (Pack-Years)	Investigated Outcomes	Jadad Score
Armitage et al., 2021 [48]	ANZCTR12614000731695	Phase I, monocentric, non-randomized, non-controlled, open-label, pilot study	4	9	Allogeneic BM-MSC (2 × 10^6^ cells/kg)	Two IV infusions, 1 week apart (1st infusion composed of radiolabelled cells, the 2nd infusion used unlabelled cells)	Stable COPD	70.0	44.0	37.0	0.0	32.0	#	NC
Squassoni et al., 2021 [56]	NCT02412332	Phase I, monocentric, randomized, open-controlled, parallel-group study	52	20	BMMC (1 × 10^8^ cells/30 mL), ADSC (1 × 10^8^ cells/30 mL), or co-administration of BMMC and ADSC (5 × 10^7^ and 5 × 10^7^ cells/30 mL)	Single IV infusion	Moderate-to-severe COPD (GOLD grade III; FEV_1_ > 30% and ≤50% predicted; no tobacco use for ≥6 months)	62.4	25.0	NA	NA	NA	Lung function and blood gas analysis	3
NA, 2021 [51]	NCT03040674	Observational, monocentric, prospective, cohort study	12	175	Autologous PBSC or co-administration of PBSC and BM-MSC	Single infusion in peripheral circulation	COPD and ILD (no active infection, no history of cancer within past 5 years)	≥16.0	64.6	34.2	0.0	NA	Lung function	NC
Harrell et al., 2020 [46]	NA	Monocentric, non-randomized, non-controlled, open-label study	3	30	PL-MSC-derived Exo-d-MAPPS product (0.5 mL), containing a high concentration of immunosuppressive factors including soluble TNF receptors I and II, IL−1 receptor antagonist, and sRAGE	One inhalation per week	COPD (post-bronchodilator FEV_1_ ≥ 30% and <80% predicted and FEV_1_/FVC < 0.7)	50.0–75.0	66.6	NA	NA	≥10.0	Lung function and exercise capacity	NC
Le Thi Bich et al., 2020 [47]	ISRCTN70443938	Monocentric, non-randomized, non-controlled, open-label, pilot study	26	20	Allogeneic UC-MSC (1.5 × 10^6^ cells/kg)	Single IV infusion	Moderate-to-severe COPD (GOLD stage C and D; post-bronchodilator FEV_1_ between 30% and 70% predicted and FEV_1_/FVC < 0.7)	67.0	100.0	NA	0.0	17.5	Lung function, number of exacerbations and exercise capacity	NC
Karaoz et al., 2020 [55]	NA	Phase I/II, monocentric, non-randomized, non-controlled, open-label study	6	5	UC-MSC (1–2 × 10^6^ cells/kg)	Four IV infusions at 2-week intervals	COPD	56.0	100.0	NA	0.0	NA	Lung function and exercise capacity	NC
Armitage et al., 2018 [52]	ANZCTR12614000731695	Phase I, monocentric, non-randomized, non-controlled, open-label, pilot study	4	9	Allogeneic BM-MSC (2 × 10^6^ cells/kg)	Two IV infusions, 1 week apart (1st infusion composed of radiolabelled cells, the 2nd infusion used unlabelled cells)	Mild-to-very-severe stable COPD (GOLD stage I, II, III, IV; no exacerbations for ≥3 months)	70.0	44.0	37.0	0.0	32.0	Lung function and hospitalization due to AECOPD	NC
Comella et al., 2017 [49]	NCT02041000	Phase I, non-randomized, open-label study	52	12	Autologous ADSC administered as SVF (1.5–3 × 10^8^ cells)	IV infusion	Severe COPD (GOLD stage III or IV; post-bronchodilator FEV_1_ ≤ 49% predicted and FEV_1_/FVC < 0.7; no active infection and/or malignancy; no current use of tobacco)	69.0	50.0	NA	0.0	NA	#	NC
De Oliveira et al., 2017 [57]	NCT01872624	Phase I, prospective, monocentric, randomized, patient-blinded, PCB (vehicle)-controlled, parallel-group study	12	10	Allogeneic BM-MSC (1 × 10^8^ cells/30 mL) +EBV	Bronchoscopical infusion in region where EBV were to be placed	Severe heterogenous pulmonary emphysema (GOLD stage III, IV; post-bronchodilator FEV_1_ < 45% predicted and FEV_1_/FVC < 0.7; no tobacco use for ≥6 months; mMRC Dyspnea Scale stage ≥ 2)	60.5	50.0	NA	NA	62.9	Lung function, hospitalization due to AECOPD, exercise capacity, and blood gas analysis	3
Rubio et al., 2017 [59]	NCT03044431	Observational, monocentric, prospective, cohort study	26	5	Autologous PBSC or co-administration of PBSC and BM-MSC	Single infusion in peripheral circulation	COPD and ILD (no active infection, no history of cancer within past 5 years)	≥16.0	54.1	36.9 (only COPD patients)	0.0	NA	Lung function	NC
Stolk et al., 2016 [53]	NCT01306513	Phase I, monocentric, prospective, non-randomized, non-controlled, open-label study	12	7	Autologous BM-MSC (1–2 × 10^6^ cells/kg) +LVRS	Two IV infusions, 1 week apart following LVRS	Severe pulmonary emphysema in both upper lung lobes (FEV_1_ ≤ 40% predicted; no tobacco use for ≥6 months)	52.4	28.6	31.4	NA	NA	Lung function	NC
Stessuk et al., 2013 [50]	NCT01110252	Follow-up of a previous Phase I study [54]	Up to 156	3	Autologous BMMC (30 mL of approximately 1 × 10^8^ cells/kg)	Single IV infusion	Severe COPD with advanced pulmonary emphysema (limited life expectancy, ineffective clinical treatments; smoking cessation for ≥6 months; mMRC Dyspnea Scale Stage > 3)	65.8	100.0	NA	NA	NA	#	NC
Weiss et al., 2013 [58]	NCT00683722	Phase II, multicenter, prospective, randomized, double-blind, PCB (vehicle)-controlled study	104	62	Allogeneic BM-MSC (Prochymal™, 100 × 10^6^ cells)	Four monthly IV infusions	Moderate-to-severe COPD (GOLD stage II, III; post-bronchodilator FEV_1_ > 30% and <70% predicted and FEV_1_/FVC < 0.7)	66.1	58.0	NA	27.1	21.5	Lung function and hospitalization due to AECOPD	4
Ribeiro-Paes et al., 2011 [54]	NCT01110252	Phase I, monocentric, non-randomized, non-controlled, open-label study	52	4	Autologous BMMC (30 mL diluted in physiological serum at 5% albumin)	Single IV infusion	Severe COPD with advanced pulmonary emphysema (limited life expectancy, ineffective clinical treatments; smoking cessation for ≥6 months; mMRC Dyspnea Scale Stage > 3)	65.8	100.0	NA	NA	NA	Lung function and blood gas analysis	NC

# Study included only in qualitative synthesis. ADSC: adipose-derived stem cells; AECOPD: acute exacerbation of COPD; BMMC: bone marrow mononuclear cells; BM-MSC: bone marrow-derived mesenchymal stem cells; COPD: chronic obstructive pulmonary disorder; EBV: endobronchial valves; Exo-d-MAPPS: Exosome-derived Multiple Allogeneic Protein Paracrine Signaling; FEV_1_: forced expiratory volume in the 1st second; FVC: forced vital capacity; GOLD: Global Initiative for Chronic Obstructive Lung Disease; IL: interleukin; ILD: interstitial lung disease; IV: intravenous; LVRS: lung volume reduction surgery; mMRC: Modified Medical Research Council Dyspnea Scale; NA: not available; NC: not calculable; PBMC: peripheral blood mononuclear cells; PCB: placebo; PBSC: peripheral blood stem cells; PL-MSC: placental tissue-derived mesenchymal stem cells; sRAGE: soluble receptor for advanced glycation end products; SVF: stromal vascular fraction; TNF: tumor necrosis factor; UC-MSC: umbilical cord-derived mesenchymal stem cells.

**Table 2 cells-11-01797-t002:** Ongoing clinical trials on stem cell-based regenerative therapies for COPD currently registered at ClinicalTrials.gov, accessed on 24 May 2022.

Trial Number Identifier	Trial Status	Trial Phase	Number of Enrolled Patients	Condition	Type of Stem Cell-Based Treatment (Dose)	Regimen and Route of Administration	Follow Up
NCT02348060	Unknown recruitment status	Observational study	100	COPD	Adipose-derived SVF containing ADSC (NA)	NA	1 year
NCT02645305	Unknown	Phase I/II	20	Moderate to severe COPD	Adipose-derived SVF containing ADSC + platelet-rich plasma (NA)	NA	1 year
NCT03500731	Recruiting	Phase I/II	8	IPF, emphysema or COPD	CD3/CD19 negative hematopoietic stem cells (NA)	NA	Up to 2 years
NCT03655795	Unknown	Phase I	20	COPD	Bronchial basal cells	NA	1 year
NCT01758055	Unknown	Phase I	12	Moderate to severe emphysema	Autologous BM-MSC (0.6 × 10^8^ cells)	Single dose, endobronchial	1 year
NCT04433104	Recruiting	Phase I/II	40	Moderate to severe COPD	UC-MSC (1 × 10^6^ cells/kg)	Two doses, the 2nd will be performed 3 months after the first transplantation, IV	1 year
NCT04047810	Active, not recruiting	Phase I	15	Advanced COPD	MSC (0.5–2 × 10^6^ cells/kg)	Single dose, IV	1 day
NCT04206007	Recruiting	Phase I	9	Moderate COPD	Ex vivo cultured human umbilical cord tissue-derived mesenchymal stem cells, named UMC119-06 (NA)	Single dose, IV	≈ 4 months
NCT04018729	Unknown status	Phase II/III	34	Severe emphysema	EV + BM-MSC (NA)	Single dose, endoscopic administration	6 months
NCT02946658	Active, not recruiting	Phase I/II	100	COPD	Adipose-derived SVF containing ADSC (NA)	Single dose, IV	1 year
NCT05147688	Recruiting	Phase I	20	COPD	UC-MSC (1 × 10^8^ cells)	Single dose, IV	4 year
NCT03899298	Active, not recruiting	Phase I	5000	COPD, among others	Amniotic stem cells and UC-MSC (NA) + nebulizer	IV + nebulizer inhalation	Up to 10 years

## Data Availability

The data presented in this study are available in the article.

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
