# Peer review of "Stem Cell-Based Regenerative Therapy and Derived Products in COPD: A Systematic Review and Meta-Analysis"

_cells, 2022, doi:10.3390/cells11111797_

Round 1

Reviewer 1 Report

This systematic review was extremely interesting and relevant to the special issue in question. The topic is relevant and ready for an SR to be done. The review was performed robustly at most steps, and I was particularly impressed by the attention given to selecting the correct tools for quality assessment, as well as the addition of GRADE to indicate confidence in the materials assessed. The use of a funnel plot and a table of ongoing studies with NCT references was also useful for future workers in the field. The introduction and discussion set the context clearly, and limitations inherent based on the included studies (eg heterogeneity of dose/lack of standardisation) have been nicely described. I have only some minor comments

1. Search strategy - this is hard to read in text format in the methods, I would suggest moving to a table in the supplement. Notably clinicaltrials.gov was not listed in the search section, although EU trials register was, and later appeared in the data extraction section. The table of ongoing studies included numbers from clinicaltrials.gov. I wonder if the search database list requires updating in the relevant section? Secondly the list of terms for COPD was relatively restrictive and did not include the exploded version of potentially related terms (eg emphysema). If using exploded terms, and in particular emphysema does this result in many more hits? I appreciate it is unlikely that the authors could go back and redo all steps of the selection process, but this could be mentioned as a limitation if not, and if the extra studies found were significant in number.

2. Throughout the manuscript some words that do not require a hyphen in the middle (eg regenerative) appear hyphenated. Please check and reformat.

Author Response

Reviewer 1

This systematic review was extremely interesting and relevant to the special issue in question. The topic is relevant and ready for an SR to be done. The review was performed robustly at most steps, and I was particularly impressed by the attention given to selecting the correct tools for quality assessment, as well as the addition of GRADE to indicate confidence in the materials assessed. The use of a funnel plot and a table of ongoing studies with NCT references was also useful for future workers in the field. The introduction and discussion set the context clearly, and limitations inherent based on the included studies (eg heterogeneity of dose/lack of standardisation) have been nicely described. I have only some minor comments.

Reply. We thank the reviewer for having positively revised our manuscript and for their minor comments that have helped us to improve the quality of our meta-analysis.

1. Search strategy - this is hard to read in text format in the methods, I would suggest moving to a table in the supplement. Notably clinicaltrials.gov was not listed in the search section, although EU trials register was, and later appeared in the data extraction section. The table of ongoing studies included numbers from clinicaltrials.gov. I wonder if the search database list requires updating in the relevant section? Secondly the list of terms for COPD was relatively restrictive and did not include the exploded version of potentially related terms (eg emphysema). If using exploded terms, and in particular emphysema does this result in many more hits? I appreciate it is unlikely that the authors could go back and redo all steps of the selection process, but this could be mentioned as a limitation if not, and if the extra studies found were significant in number.

Reply. As suggested, the expanded search string has been switched into the supplementary file (Table S1), by reporting also detailed translations. Only a summary and well-comprehensive version of the search string has been reported in the main manuscript. Concerning clinicaltrials.gov, probably there was a misunderstanding since we reported this repository database as the first one (it was already reported in the manuscript): “The search was performed in ClinicalTrials.gov, Cochrane Central Register of Controlled Trials (CENTRAL)…..”. We have now improved the search string and redo all the steps of the selection processes by adding also the term “emphysema”; the flowchart and date of search have been amended accordingly and, fortunately, no further studies were eligible to be included in the meta-analysis.

2. Throughout the manuscript some words that do not require a hyphen in the middle (eg regenerative) appear hyphenated. Please check and reformat.

Reply. We have amended this typographic matter that was caused by the pagination automatically performed by the platform of the Journal.

Reviewer 2 Report

This is a useful review of stem cell regenerative therapy in COPD.  The review should address how such studies should be evaluated for efficacy.  The FEV1 and six-minute walk tests are clinical endpoints which have wide variability and reproducibility.  The lung tissue destruction in COPD affects almost all structural components of the lung so the outcomes of therapy should be evaluated in the cellular and biochemical processes of the lung including biomarkers where possible. A comment in this regard seems appropriate in a review of this magnitude.

Author Response

Reviewer 2

This is a useful review of stem cell regenerative therapy in COPD.  The review should address how such studies should be evaluated for efficacy.  The FEV1 and six-minute walk tests are clinical endpoints which have wide variability and reproducibility.  The lung tissue destruction in COPD affects almost all structural components of the lung so the outcomes of therapy should be evaluated in the cellular and biochemical processes of the lung including biomarkers where possible. A comment in this regard seems appropriate in a review of this magnitude.

Reply. We thank the reviewer for having positively revised our manuscript and for their minor comments that have helped us to improve the quality of our discussion. We have amended the manuscript according to the comment by adding a section in the discussion regarding the relevance of assessing the efficacy outcomes also at cellular and biochemical levels including also potential biomarkers.